# From Weight-Based to State-Based Fine-Tuning: Further Memory Reduction on LoRA with Parallel Control

**Chi Zhang** [1]  **Lianhai Ren** [1]  **Jingpu Cheng** [1]  **Qianxiao Li** [1 2]

{czhang24, qianxiao}@nus.edu.sg;  {lianhairen, chengjingpu}@u.nus.edu

## Abstract

The LoRA method has achieved notable success in reducing GPU memory usage by applying low-rank updates to weight matrices. Yet, one simple question remains: can we push this reduction even further? Furthermore, is it possible to achieve this while reducing computation time and preserving performance? Answering these questions requires moving beyond the conventional weight-centric approach. In this paper, we present a state-based fine-tuning framework that shifts the focus from weight adaptation to optimizing forward states, with LoRA acting as a special example. Specifically, state-based tuning introduces parameterized perturbations to the states within the computational graph, allowing us to control states across an entire residual block. A key advantage of this approach is the potential to avoid storing large intermediate states in models like transformers. Empirical results across multiple architectures—including ViT, RoBERTa, LLaMA2-7B, and LLaMA3-8B—show that our method further reduces memory consumption and computation time while preserving performance. As a result of memory reduction, we explore the feasibility to train 7B/8B models on consumer-level GPUs like Nvidia 3090, without model quantization. The code is available here.

## 1. Introduction

Low-Rank Adaptation (LoRA) (Hu et al., 2021) has emerged as an effective algorithm for fine-tuning pre-trained models, demonstrating notable success across domains such as natural language processing and computer vision. At its core, LoRA leverages low-rank decomposition $\Delta W = AB$ to approximate weight updates with some low-rank matrices $A$ and $B$, allowing only a small subset of parameters to be trained. Such a design not only maximally preserves the performance of full-tuning but also makes training feasible on hardware with limited resources.

Yet, such a core explanation is challenged by recent fine-tuning approaches (Chen et al., 2022; Zhang et al., 2024b), in which the update rule often takes the form $x' = f(W, x) + g(\Delta W, x)$. Here, $f$ represents the pretrained dynamics while $g$ introduces a nonlinear perturbation based on the fine-tuned parameters. The presence of the nonlinear function $g$ precludes a straightforward interpretation of $\Delta W$ as a simple low-rank modification to the weight matrix $W$. As a result, this departure from linearity *challenges the foundational view* of LoRA as merely performing low-rank weight updates. This raises a natural question: if low-rank decomposition no longer fully captures the nature of these adaptations, what alternative frameworks might offer a more comprehensive understanding?

Another significant challenge lies in further reducing the GPU memory consumption of LoRA. While approaches like VeRA (Kopiczko et al., 2023), BinaryLoRA (Zhang et al., 2024a), and DoRA (Liu et al., 2024) have made strides by using vector-based, binary LoRAs or halving rank, memory reduction often remains limited. For example, halving the rank in DoRA reduces parameters by nearly $50\%$, but results in only a limited decrease of $0.256$ GB GPU memory on an 8B model. A closer analysis reveals that the primary bottleneck is no longer the storage of parameters, but rather the memory of maintaining forward states. This limitation becomes particularly critical as 7B/8B models become increasingly common, yet training them remains infeasible on consumer-level GPUs like Nvidia 3090/4090. This shift underscores the need for strategies to address the memory demands associated with activations during the forward pass, enabling more memory-efficient fine-tuning techniques.

Addressing these challenges requires moving beyond the traditional weight-centric paradigm. While LoRA and its

---

[1]Department of Mathematics, National University of Singapore [2]Institute for Functional Intelligent Materials, National University of Singapore. Correspondence to: Qianxiao Li <qianxiao@nus.edu.sg>.

*Proceedings of the 42$^{nd}$ International Conference on Machine Learning*, Vancouver, Canada. PMLR 267, 2025. Copyright 2025 by the author(s).

variants focus on adjusting specific layer weights within this framework, achieving further memory reductions calls for new theoretical insights and the development of alternative fine-tuning frameworks. One promising direction emerges from recent work (Zhang et al., 2024b), which draws connections between fine-tuning and control theory. In control theory, directly modifying the core parameters of a system is often impractical. For example, directly adjusting the motor settings of a robot can be difficult; instead, it is typically more effective to regulate system states—such as position or velocity—through feedback control mechanisms (Franklin et al., 2002; Kirk, 2004).

The goal of this paper is to introduce a generalized state-based fine-tuning framework that facilitates direct adjustment of network states. By representing the neural network as a directed acyclic graph (DAG), this framework enables manipulation of states connected by arbitrary edges, rather than being limited to weight parameters alone. Furthermore, the state-based approach supports the adjustment of states that are not necessarily immediate neighbors and can account for hierarchical dependencies within the network. This flexibility allows for more sophisticated control strategies over the network's behavior, extending beyond the conventional reliance on low-rank matrix updates.

Thanks to this new framework, we can design more efficient algorithms by treating entire components, such as MLP or attention blocks, as single units. A key advantage of this design is the reduction in memory usage, as it circumvents the need of storing large intermediate states associated with individual layers or operations. Moreover, by leveraging low-rank matrices, we can reduce both the parameters and the states that need to be stored, resulting in highly memory-efficient algorithms.

The proposed algorithm outperforms traditional methods across a range of tasks, while using less GPU memory and reducing computation time. These advantages are demonstrated across several model architectures, including ViT (Dosovitskiy et al., 2020), RoBERTa (Liu, 2019), LLaMA2-7B (Touvron et al., 2023), and LLaMA3-8B (Dubey et al., 2024). Additionally, by further reducing memory usage, we demonstrate the feasibility of training 7B/8B models on a Nvidia 3090 GPU with 24GB memory, without resorting to model quantization. Our results reveal that, even with lower-memory GPUs, the proposed approach achieves performance comparable to setups with higher memory capacity. This makes the approach accessible to a wide range of researchers and practitioners with only consumer-level GPUs.

In summary, our contributions are four-fold: (1) We introduce a state-based fine-tuning approach that shifts the focus from weight-centric adaptation to optimizing the model's forward states, providing a more general framework and greater flexibility in designing parameter-efficient algorithms. (2) Building on this framework, we demonstrate that memory usage can be further reduced by applying low-rank perturbations to entire components, such as MLP and attention blocks, thus avoiding the need to store large intermediate states. (3) Empirical results show that our approach achieves superior performance while simultaneously reducing GPU memory usage and computation time. (4) We explore the possibility of training large models on consumer-level GPUs, allowing more researchers/practitioners to benefit from PEFT studies.

## 2. Related Works

**Parameter-Efficient Fine-Tuning**  Owing to the large parameter size of modern transformer-based models (Vaswani et al., 2017; Dosovitskiy et al., 2020), fully tuning these models becomes challenging. Early works (Oquab et al., 2014; Alain, 2016) in transfer learning focused on linear probing, where only the final layer is fine-tuned. These approaches have been extended to selectively tuning the middle or bottom layers in (Lee et al., 2022; Lodha et al., 2023; Kaplun et al., 2023; Nagae et al., 2022). Recent parameter-efficient fine-tuning (PEFT) methods propose to inject new trainable parameters, while keeping the pretrained model fixed. In particular, Adapter methods and their variants (Houlsby et al., 2019; Rebuffi et al., 2017; Karimi Mahabadi et al., 2021) employ bottleneck structures inserted after self-attention and feed-forward network layers to achieve parameter efficiency. Prompt-based methods such as Prompt Tuning (Lester et al., 2021), P-Tuning (Liu et al., 2021), and Prefix-Tuning (Li & Liang, 2021) optimize trainable prompts rather than tuning the weights of the model. LoRA (Hu et al., 2021) is among the most popular PEFT methods. It leverages the idea of low-rank decomposition for the updated weights, significantly reducing the number of trainable parameters. Variants of LoRA (Zhang et al., 2023a; Kopiczko et al., 2023; Zhang et al., 2023b) further optimize efficiency by freezing or sharing the LoRA matrices, reducing memory requirements through quantization techniques (Dettmers et al., 2023), or using an adaptive strategy. A recent extension, Dora (Liu et al., 2024), decomposes the magnitude and direction of the low-rank weight matrices, offering improved performance. However, LoRA and these variants remain weight-centric, in comparison to the state-based tuning proposed in this paper.

**Control in Machine Learning**  Control theory (Kirk, 2004) focuses on managing dynamical systems by developing models or algorithms that govern system inputs to drive the system toward desired states. From a dynamical perspective, residual neural networks can be viewed as dynamical systems, with the model representing the system and the layer inputs/outputs serving as the states (E, 2017;

Li et al., 2017). This connection between deep learning and control theory has inspired numerous advancements in various domains, including approximation theory (Tabuada & Gharesifard, 2020; Li et al., 2022; Cheng et al., 2023), network structure design (Haber & Ruthotto, 2017; Lu et al., 2017; Nguyen et al., 2024), and optimizer development (Li & Hao, 2018; Benning et al., 2019). The parameter-efficient fine-tuning task can naturally be formulated as a controlled dynamical system (Zhang et al., 2024b), where the original model represents the system, and controllers are designed to adapt the system to downstream tasks. Our work provides a more general state-based tuning framework, where this work can be considered as a special case of choosing the control function $g$.

## 3. A State-Based Fine-Tuning Framework

### 3.1. Weight-Based Fine-Tuning (Weight-FT)

We begin by introducing the Low-Rank Adaptation (LoRA) method (Hu et al., 2021), which facilitates parameter-efficient fine-tuning through low-rank decomposition. For a pretrained weight-matrix $W_0 \in \mathbb{R}^{d \times k}$, LoRA models the weight update $\Delta W \in \mathbb{R}^{d \times k}$ with a low-rank decomposition $\Delta W = AB$, where $A \in \mathbb{R}^{d \times r}$ and $B \in \mathbb{R}^{r \times k}$ represent two low-rank matrices with rank $r \ll \min(d, k)$. Consequently, the fine-tuned weight $W'$ becomes:

$$W' = W_0 + \Delta W = W_0 + AB, \qquad (1)$$

where $W_0$ remains fixed during the fine-tuning process, and only the low rank matrices $A, B$ are updated.

Following LoRA, many PEFT algorithms further explore and extend such a weight decomposition mechanism, including VeRA (Kopiczko et al., 2023) and DoRA (Liu et al., 2024) to utilize alternative low-rank structures, EVA (Paischer et al., 2024) to introduce data-driven initialization, and rsLoRA (Kalajdzievski, 2023) to stabilize LoRA.

In broader terms, these approaches can be categorized as weight-based fine-tuning (Weight-FT) algorithms, as they work on modifying the model's weight matrices to approximate the low-rank changes during the fine-tuning process. The fundamental hypothesis behind these weight-FT algorithms is that fine-tuning updates tend to have a low "intrinsic rank" (Aghajanyan et al., 2020; Hu et al., 2021). Consequently, low-rank matrices can be employed to efficiently adjust the pretrained weights.

But modern fine-tuning strategies (Chen et al., 2022; Zhang et al., 2024b) have introduced alternative methods, many of which go beyond this weight-FT paradigm. For instance, a pretrained layer can be fine-tuned using either of the following formulations:

$$x_{t+1} = x_t + f_t\left((W_t + \Delta W_t)x_t\right), \qquad (2)$$

$$x_{t+1} = x_t + f_t\left(W_t x_t\right) + \Delta W_t x_t, \qquad (3)$$

where $f_t$ represents nonlinear functions, such as ReLU or GeLU. Notably, the latter formulation cannot be interpreted as weight adjustment like LoRA, as the presence of a nonlinear function makes it difficult to explain through the lens of weight decomposition. This limitation highlights the need for a more general fine-tuning framework that can accommodate a broader range of strategies, including those involving nonlinear transformations.

### 3.2. Inspirations from Control Theory

To proceed, we need to move beyond these classical weight-centric methodologies. In particular, the pioneering work (Zhang et al., 2024b) has demonstrated that Eq (3) align closely with the classical control problem. Consider the continuous-time ordinary differential equation (ODE) with affine control (Franklin et al., 2002):

$$\dot{x}_t = f_t\left(W(t)x(t)\right) + G(t)u(t). \qquad (4)$$

In state-feedback control systems, the control signal $u(t)$ is typically defined as a function of the state $x(t)$ through a feedback gain matrix $K(t)$, such that $u(t) = K(t)x(t)$. Substituting $u(t)$ into the original system gives:

$$\dot{x}_t = f_t\left(W(t)x(t)\right) + G(t)K(t)x(t). \qquad (5)$$

The central connection lies in the role of the additive terms: the control term $G(t)K(t)x(t)$ in the continuous system (5) modifies the state trajectory by applying a control matrix $G(t)$ and a feedback gain $K(t)$. Analogously, Eq (3) can be interpreted as a discretized extension of Eq (5), where $\Delta W_t x_t$ functions as a control-inspired mechanism that influences state updates.

### 3.3. State-Based Fine-Tuning (State-FT)

The key insight from the above connection is that these updates primarily target the *system's states* $\{x_t\}$, rather than the parameters governing them. Therefore, in this part, we propose a state-based fine-tuning (State-FT) framework that shifts the focus from parameter updates to directly influencing the system's states.

Formally, consider a neural network as a directed acyclic graph $G = (V, E)$, where each node $v \in V$ represents a computational state, and each directed edge $(u, v) \in E$ represents a transformation applied to the state at node $u$, producing the state at node $v$.

In particular, the computation on the edge $(u, v) \in E$ is defined as:

$$x_v^u = f_v^u(x_u; W_{u \to v}),$$

where $W_{u \to v}$ represents the weight matrix for the transformation along edge $(u, v)$, and $f_v^u$ is the function applied, such as a linear transformation or nonlinear activation.

State-based tuning involves directly modifying the intermediate states $x_v$ in the graph. For example, if an edge $(u, v) \in E$ is selected for fine-tuning, then the updated state is defined as:

$$x'_v = \sum_{\tilde{u} \in \mathcal{A}(v)} x_v^{\tilde{u}} + \Delta x_v^u,$$

where $\mathcal{A}(v) = \{\tilde{u} \mid \tilde{u} \text{ is an ancestor of } v\}$ denotes the ancestors of node $v$ and $\Delta x_v^u$ represents the adjustment from the a state $x_u$. This adjustment can be parameterized as:

$$\Delta x_v^u = g_v^u(M_{u \to v}, x_u),$$

where $g_v^u$ is the control function, and $M_{u \to v}$ is a learnable control matrix to modify the states.

The full update for $x_v$ then becomes:

$$x'_v = \sum_{\tilde{u} \in \mathcal{A}(v)} f_v^{\tilde{u}}(x_{\tilde{u}}; W_{\tilde{u} \to v}) + g_v^u(M_{u \to v}, x_u). \quad (6)$$

For simplicity, we will use $f_v^u$ in place of $f_v^{\tilde{u}}$ whenever no confusion arises in the following parts.

The above state-based fine-tuning strategy fundamentally differs from weight-based approaches by focusing on the direct manipulation of intermediate states rather than adjusting the model's weight matrices. While LoRA uses low-rank matrices to update weights, state-based fine-tuning involves modifying a state pair $(x_u, x_v)$. This shift allows for more flexible and granular control over the model's behavior, as the state dynamics are influenced by control functions on the system's states. Additionally, state-based fine-tuning can incorporate more complex, non-linear transformations, unlike LoRA, which is limited to linear perturbations of the weight matrices. As a result, the state-based framework provides a broader range of adaptation strategies.

**Revisiting LoRA as a Special Case**  We demonstrate that LoRA can be viewed as a special case of the proposed framework by choosing specific forms for $f_v^u$ and $g_v^u$. Specifically, LoRA chooses specific edges $(u, v) \in E$ with a linear layer,

$$f_v^u(W_{u \to v}, x_u) = W_{u \to v} x_u,$$

such as the query (Q), key (K), and value (V) blocks. The control function $g_v^u$ is defined as a low-rank update on the weight matrix:

$$g_v^u(B_{u \to v}, x_u) := A_{u \to v} B_{u \to v} x_u, \quad (7)$$

where $A_{u \to v}$ and $B_{u \to v}$ are learnable low-rank matrices, used to update the pre-existing weight matrix $W_{u \to v}$. As such, from the perspective of State-FT, the LoRA algorithm can be seen as a special case in which the selected edges $(u, v)$ are restricted to direct neighbors associated with linear layers.

**More General Choice of $f_v^u$ and $g_v^u$**  More generally, the function $f_v^u$ can involve more complex transformations, such as those defined by multiple layers or non-linear mappings that go beyond simple weight matrices. Similarly, the control function $g_v^u$ can also be more intricate, potentially incorporating more layers, non-linearities, or even dynamic control mechanisms. For example, assume $f_v^u$ consists of two linear layers followed by a non-linearity:

$$f_v^u(W_{u \to v}, x_u) = \sigma_2\left(W_2\left(\sigma_1(W_1 x_u + b_1)\right) + b_2\right),$$

and keep the control function $g_v^u$ as low-rank matrices in Eq (7). As such, we are able to control multiple layers with a parameter-efficient control strategy. This generalization allows for more flexibility in the state-based tuning process, enabling it to model more complex, hierarchical transformations of states during fine-tuning.

## 4. Further Memory Reduction on LoRA with State-FT

The flexibility of choosing arbitrary $(u, v) \in E$ within the state-FT framework allows for more general and versatile approaches to model adaptation. For example, we may allow the state $x_v$ to receive multiple updates $\Delta x_v^u$, to use weight differences $x_v - x_u$, akin to classical PID control (Åström & Hägglund, 2006). In this section, we present an example to design new parameter-efficient tuning algorithms, primarily aligned with the PEFT framework, to further reduce LoRA's memory requirements.

### 4.1. State-FT on a Complete Residual Block

Modern neural networks are commonly composed of multiple blocks, each structured with a residual connection (He et al., 2016). To align with this paradigm, we consider a specific case of the State-FT where layers connected by a shortcut (or residual connection) are grouped into one computational block $f_v^u$. By doing so, we are able to leverage the inherent structure of these networks and preserve the integrity of the residual connections.

Formally, for a block of layers connected by a shortcut, let the input to the block be $x_u$, and the output be $x_v$. The computation for the block can be expressed as:

$$x_v = f_v^u(x_u; \{W_{u \to v}\}) + x_u,$$

where $\{W_{u \to v}\}$ represents the collection of weight matrices for all layers within the block and $f_v^u$ aggregates the transformations applied within the block.

In the State-FT framework, the updated state for the block incorporates both the transformations and a state adjustment term, defined as:

$$x'_v = x_u + \mathcal{F}_v(x_u; \{W_{u \to v}, M_{u \to v}\}),$$

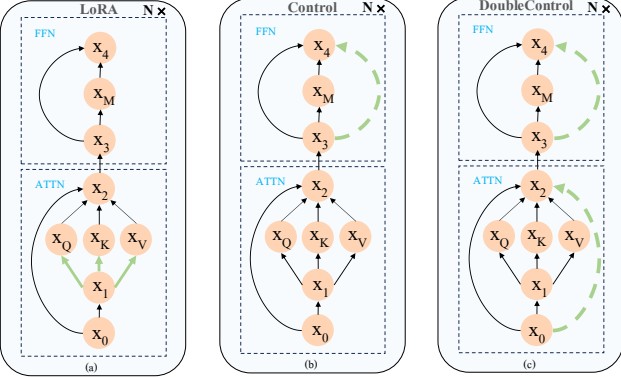

Figure 1: Figure (a) illustrates the computational graph in LoRA, where specific edges (Q, K, V) with linear transformations are selected for tuning. Figure (b) introduces the control approach, where a new edge (in green color) across the FFN block is introduced. Figure (c) extends this control to both the attention and FFN blocks, resulting in the double-control method.

where $\mathcal{F}_v(\cdot)$ is a composite functions defined as:

$$\mathcal{F}_v(x_u; \{W_{u \to v}, M_{u \to v}\}) = f_v^u(x_u; W_{u \to v}) + g_v^u(M_{u \to v}; x_u).$$

Here $g_v^u(M_{u \to v}; x_u)$ is the control function that modifies the state using the control matrix $M_{u \to v}$. The choice of $g_v^u$ can significantly influence the behavior and adaptability of the model. For instance, $g_v^u$ can incorporate more complex architectures, such as multi-layer perceptrons (MLPs) or non-linear layers, to act as universal function approximators. Alternatively, $g_v^u$ can be designed as a simple low-rank matrix transformation to reduce trainable parameters and memory footprint.

Consider the transformer architecture. We are now free to pick the starting and ending state of the residual block as an edge to tune. One option is to choose the starting and ending nodes of the feed-forward network (FFN) block, which typically includes a LayerNorm (LN) (Ba, 2016) and multiple linear layers. We refer to this method as the "Parallel Control" approach in Figure 1(b). Alternatively, we can extend this strategy to fine-tune both the attention block and the FFN block within the residual structure, leading to what we term the "Double Control" approach. Note that this differs from classical LoRA, which selects specific edges with linear transformations to adjust (e.g., Q, K, V in Figure 1(a)).

### 4.2. Reducing Memory by Releasing Intermediate States

A fundamental question is why we aim to compress the residual block into a single computational unit. To answer this, Figure 2 presents a simple example involving two MLP layers to illustrate GPU memory consumption during fine-

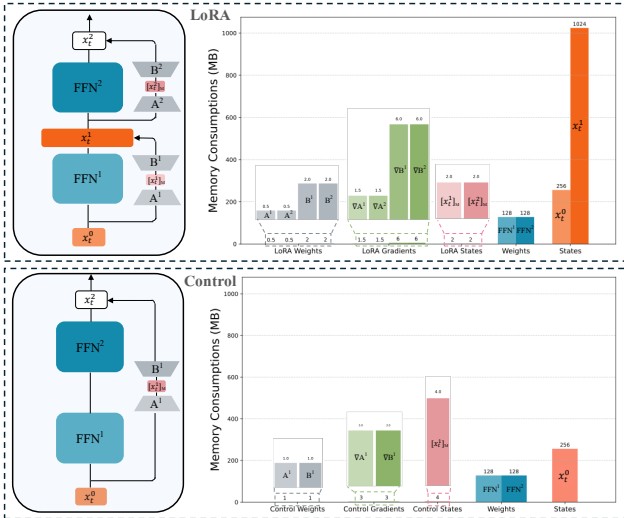

Figure 2: Memory consumption analysis for LoRA (top) and Parallel Control (bottom) on two MLP layers. In LoRA, the dominant memory usage comes from the intermediate state $x_t^1$. By treating the MLP layers as a single unit and bypassing the intermediate states, the Parallel Control significantly reduces memory requirements.

tuning. Consider a typical fine-tuning scenario with a batch size of 16, a token length of 1024, and a feature dimension of 4096. Thanks to the foundational LoRA framework, the trainable parameters $A$, $B$, and their gradients require only 5 MB and 15 MB of GPU memory, respectively. The dominant memory consumption now comes from the forward states $x_t$ and the model weights $W_t$. Specifically, the intermediate state $x_t^1$ contributes the largest portion of memory usage (65.24%), due to the large feature size (16,384 per token).

Storing this intermediate state is essential in the LoRA algorithm, as it is needed for the later gradient computation of $A_t^2$ w.r.t. the final loss $\ell$:

$$\frac{\partial \ell}{\partial A_t^2} = \frac{\partial \ell}{\partial x_t^2} \frac{\partial x_t^2}{\partial [x_t^2]_M} \frac{\partial [x_t^2]_M}{\partial A_t^2}, \qquad (8)$$

where $\frac{\partial [x_t^2]_M}{\partial A_t^2} = x_t^1$. As such, computing $\frac{\partial \ell}{\partial A_t^2}$ would require the system to store $x_t^1$ in the forward pass. Similarly, the state $x_t^0$ needs to be stored, although its size is much smaller.

In contrast, when multiple MLP layers are treated as a single unit, this large intermediate state $x_t^1$ can be released in memory. This is achieved by using the parallel scheme in Figure 2 (bottom left), which bypasses this intermediate state. Consequently, only the comparatively smaller state $x_t^0$ needs to be stored, resulting in a substantial reduction in GPU memory consumption.

The above example demonstrates why compressing multiple

layers into one unit $f_u^v$ could potentially save GPU memory. Note the reduction becomes more significant as the batch size increases, the feature length expands, or multiple linear layers are stacked together. In such cases, the memory savings provided by the proposed approach are more pronounced, enabling the system to handle larger workloads while keeping memory consumption in check.

### 4.3. Reducing Additional Parameter Numbers

In addition to reducing the memory, we show the proposed control mechanism can also reduce the overall parameters or adopt a larger rank with the same parameter budget.

Injecting new low-rank matrices would nevertheless introduce new parameters, as well as additional intermediate states. For a given state $x_t \in \mathbb{R}^{m \times d}$, the standard FFN block of a transformer architecture first expands the feature to $4d$ and then reduces back to $d$. Applying the LoRA algorithm to these layers involves introducing low-rank matrices $A$ and $B$ with dimensions $d \times r$ and $r \times 4d$. Consequently, tuning the FFN block requires a total of $10rd$ parameters.

In contrast, within the state-FT framework, multiple MLP layers and the corresponding LayerNorm can be viewed as a single unit $f_v^u$, enabling the application of a unified update across the entire block. By applying the low-rank design to the control function $g_v^u(M_{u \to v}, x_u)$, the update requires only two matrices of dimensions $d \times r$ and $r \times d$, resulting in a total of only $2rd$ parameters. This approach significantly reduces parameter requirements by bypassing the feature expansion stage in the FFN block. Alternatively, it allows for the adoption of a higher rank within the same parameter budget, thereby enhancing the effectiveness of fine-tuning.

### 4.4. Performance Analysis

Yet, we have to ensure that the reduction in memory and parameters does not compromise overall performance. To validate this, we present an analysis of the expressive power of the proposed parallel control method in this part.

For the simplest case of deep linear networks, we have the following result:

**Theorem 4.1.** *Consider a deep linear network defined as:*

$$f : x_0 \to x_T, \quad x_{t+1} = W_t x_t, \quad t = 0, \dots, T, \quad (9)$$

*and its low-rank adaptation:*

$$\bar{f} : x_0 \to x_T, \quad x_{t+1} = (W_t + R_t) x_t, \quad t = 0, \dots, T-1, \quad (10)$$

*where $x_t \in \mathbb{R}^{d_t}$ represents the hidden state, $W_t \in \mathbb{R}^{d_{t+1} \times d_t}$ is the weight matrix at layer $t$, and $R_t \in \mathbb{R}^{d_{t+1} \times d_t}$ is a low-rank matrix with rank $r_t$.*

*Then, there exists a weight matrix $M$ satisfying*

$$\mathrm{rank}(M) \leqslant r_0 + \cdots + r_{T-1}, \quad (11)$$

*such that for all $x_0 \in \mathbb{R}^{d_0}$,*

$$\bar{f}(x_0) = f(x_0) + M x_0. \quad (12)$$

This result implies that in the case of deep linear networks, the parallel control method can always achieve the *same expressive power* as LoRA, provided they share the same total rank. Consequently, for linear models, it is always safe to use parallel control without loss of expressive capability.

For non-linear blocks, the following result offers insights into the potential advantages of parallel control over a single state.

**Theorem 4.2.** *Let $F_{x_t}$ and $G_{x_t}$ be the mappings $\Delta W_t \mapsto x_{t+1} \in \mathbb{R}^d$, as defined in equations* (2) *and* (3)*, respectively. If $\nabla f(x_t)$ is singular, then the pushforward of the tangent space at $0$ under $F_{x_t}$ forms a proper subspace of $\mathbb{R}^d$. In contrast, the pushforward of the tangent space at $0$ under $G_{x_t}$ always spans the entire space $\mathbb{R}^d$, as long as $x_t$ is non-zero.*

This theorem suggests that when $\nabla f(x_t)$ is singular, local perturbations in $x_t$ under LoRA are restricted to a subspace, limiting control over the state. In contrast, parallel control remains unaffected. This insight highlights that parallel control can potentially provide *greater adaptability* in the presence of degeneracies in the original model.

### 4.5. Rethinking Parallel Control: Affine vs Non-Affine Control Structures

In fact, the difference between equations (2) and (3) extends beyond a superficial rearrangement of terms or a simple modification of the computational graph. Instead, these formulations correspond to fundamentally distinct system structures when viewed through the lens of control theory.

Equation (2), which represents the classical LoRA formulation, can be seen as a *non-affine control system*,

$$\dot{x}(t) = f(x(t), u(t)),$$

where the control input $u(t)$, or the trainable matrix $\Delta W_t$ in LoRA, is embedded inside a nonlinear transformation. This makes the system nonlinear in both the state $x_t$ and the control $\Delta W_t$. Analyzing such a non-affine control system often tends to be complicated.

In contrast, the "Parallel Control" proposed in this work follows a *control-affine structure* of the form:

$$\dot{x}(t) = f(x(t)) + x(t)u(t),$$

where the control component is decoupled from the function $f$. This decomposition separates the dynamics into a nominal term and an additive control perturbation. From a learning perspective, this distinction has important implications.

Affine and non-affine controls represent fundamentally different classes of control design, each with unique stability and efficiency properties. Affine-control systems, in particular, are well-studied in classical and modern control theory, particularly in feedback and optimal control (Franklin et al., 2002; Skaf & Boyd, 2010; Goswami & Paley, 2021; Li et al., 2023). This structure also aligns naturally with modular fine-tuning strategies, allowing adaptation mechanisms to operate independently of the nonlinear core dynamics $f(\cdot)$. In contrast, general nonlinear (non-affine) control systems like LoRA and its variants embed the control input within nonlinear transformations, making analysis and design significantly more complex and often intractable.

In summary, the proposed formulation not only introduces a new edge in the computational graph, but also fundamentally redefines the system structure in a way that enhances control interpretability and potential theoretical analysis. By adopting a control-affine perspective, it opens the door to simpler and interpretable analysis, paving the way for future developments in fine-tuning theory and practice.

# 5. Experiments

In this section, we evaluate the effectiveness of the proposed State-FT parallel control methods, compared to traditional weight-FT approaches such as LoRA (Hu et al., 2021) and DoRA (Liu et al., 2024). Our experiments cover a diverse range of model architectures, including ViT, RoBERTa, LLaMA2-7B and LLaMA3-8B.

## 5.1. A Toy Example on Vision Transformer (ViT)

We begin our evaluation with a toy example using the CIFAR-100 dataset (Krizhevsky et al., 2009) on the ViT model (Dosovitskiy et al., 2020). The objective of this experiment is to demonstrate the feasibility of treating multiple components as a single unit within the State-FT framework. Specifically, we extend the FFN block by adding two additional MLP layers after the activation function. The State-FT framework enables us to treat the LayerNorm and all MLP layers as a unified function $f_v^u$, in contrast to classical LoRA algorithms, which apply low-rank decomposition separately to each layer's weights. To ensure a fair comparison, both methods are configured to use an equal number of trainable parameters, and fine-tuned for 50 epochs.

Table 1: Comparison of algorithms on the extended ViT-B model. [†] Training time is tested on a single Nvidia-3090.

| Algorithm | # of Params | GPU Memory↓ | Training Time↓[†] | Accuracy↑ |
|---|---|---|---|---|
| LoRA | 1.27 M | 18.010 GB | 4h 42m | $91.84_{\pm 0.11}$ |
| Control | 1.27 M | **12.280 GB** | **3h 24m** | $\mathbf{91.96}_{\pm 0.05}$ |

Tuning each layer individually, as in the classical LoRA approach, requires the storage of multiple intermediate states during the forward pass, resulting in significantly higher GPU memory usage. Specifically, LoRA's memory consumption is approximately $46.66\%$ greater than that of the parallel control method, which also results in a $38.24\%$ increase in training time. Despite these gains in memory efficiency and reduced training time, the parallel control approach obtains similar accuracy performance as LoRA. Therefore, the parallel control approach proves to be an effective fine-tuning method, offering lower memory consumption and reduced training time while maintaining competitive performance.

## 5.2. GLUE Benchmark with RoBERTa Models

We further apply the parallel control approach to the GLUE benchmark (Wang, 2018), where the experiment settings follow the original LoRA paper (Hu et al., 2021). The experimental parameters, including the choice of pretrained model, learning rate, and ranks for the LoRA weights, are consistent with those used in the LoRA paper. Details of these settings are available in Appendix B.1. But we also make a few changes. (1) Unlike the original LoRA paper, which first trains models on MNLI and selects the best checkpoint, we omit this step to reduce complexity and improve reproducibility, especially for datasets with limited samples like MRPC, RTE, and STS-B, as suggested by (Wu et al., 2024). (2) We fine-tune the entire RoBERTa head, which includes multiple MLP layers, instead of just the last layer. While this increases the number of parameters (Table 3), it generally results in more stable performance.

The results, summarized in Table 2, highlight the performance of the proposed Control method compared to LoRA and DoRA across eight GLUE benchmark tasks. Across these tasks, the Control approach demonstrates superior performance on seven out of eight. For tasks with sufficient samples (e.g., MNLI), the improvement is marginal. However, on other tasks, such as CoLA, the Control method outperforms LoRA by $2.05\%$ and DoRA by $1.68\%$. This improvement demonstrates the method's ability to handle more complex linguistic patterns effectively. The only exception is the STS-B dataset, where the Control method is marginally outperformed by DoRA on RoBERTa-base by $0.19\%$.

The more important parts are the reduced memory and computation time. Table 3 reports the number of parameters, GPU memory usage, and training time for RoBERTa-base on the CoLA dataset (Warstadt, 2019). The control approach demonstrates the lowest GPU memory usage, requiring only $12.634$ GB, which is $1.396$ GB less than LoRA and $3.652$ GB less than DoRA. Furthermore, the control approach achieves the shortest training time, completing in 4 minutes and 59 seconds, making it approximately 10.2% faster than

Table 2: Comparison of algorithm performance on the GLUE benchmark. We report the overall (matched and mismatched) accuracy for MNLI, Matthew's correlation for CoLA, Pearson correlation for STS-B, and accuracy for other tasks.

| Model & Method | MNLI | SST-2 | MRPC | CoLA | QNLI | QQP | RTE | STS-B | Avg |
|---|---|---|---|---|---|---|---|---|---|
| $Rob_{base}$ (LoRA) | $87.53_{+0.08}$ | $95.11_{+0.23}$ | $88.64_{+0.23}$ | $63.27_{+0.81}$ | $93.06_{+0.15}$ | $90.79_{+0.04}$ | $75.79_{+1.57}$ | $90.74_{+0.09}$ | 85.62 |
| $Rob_{base}$ (DoRA) | $86.28_{+0.13}$ | $94.84_{+0.09}$ | $88.40_{+0.11}$ | $63.64_{+0.33}$ | $92.96_{+0.07}$ | $90.11_{+0.05}$ | $76.05_{+1.67}$ | $\mathbf{90.78}_{+0.08}$ | 85.38 |
| $Rob_{base}$ (Control) | $\mathbf{87.57}_{+0.16}$ | $\mathbf{95.26}_{+0.06}$ | $\mathbf{89.30}_{+0.31}$ | $\mathbf{65.32}_{+0.44}$ | $\mathbf{93.12}_{+0.04}$ | $\mathbf{91.04}_{+0.02}$ | $\mathbf{76.89}_{+0.78}$ | $90.59_{+0.15}$ | $\mathbf{86.14}$ |
| $Rob_{large}$ (LoRA) | $90.89_{+0.12}$ | $96.56_{+0.19}$ | $90.52_{+0.30}$ | $68.13_{+0.12}$ | $95.13_{+0.08}$ | $91.79_{+0.06}$ | $84.48_{+0.29}$ | $92.00_{+0.16}$ | 88.69 |
| $Rob_{large}$ (DoRA) | $90.03_{+0.33}$ | $96.63_{+0.19}$ | $90.85_{+0.24}$ | $68.82_{+0.03}$ | $95.15_{+0.07}$ | $90.25_{+0.05}$ | $84.72_{+0.85}$ | $\mathbf{92.45}_{+0.13}$ | 88.61 |
| $Rob_{large}$ (Control) | $\mathbf{90.91}_{+0.11}$ | $\mathbf{96.67}_{+0.25}$ | $\mathbf{90.94}_{+1.23}$ | $\mathbf{69.85}_{+0.31}$ | $\mathbf{95.21}_{+0.10}$ | $\mathbf{91.94}_{+0.04}$ | $\mathbf{86.28}_{+0.36}$ | $\mathbf{92.45}_{+0.10}$ | $\mathbf{89.28}$ |

Table 3: Number of parameters, GPU memory and the training time (on a single Nvidia-3090) for RoBERTa base on the RTE dataset. Batch size is 256, and epoch is 80.

| Algorithm | # of Params | GPU Memory↓ | Training Time↓ |
|---|---|---|---|
| LoRA | 0.88 M | 14.030 GB | 5m 33s |
| DoRA | 0.91 M | 16.286 GB | 7m 10m |
| Control | 0.88 M | **12.634 GB** | **4m 59s** |

LoRA and 30.4% faster than DoRA.

Finally, the control approach primarily focuses on tuning the MLP layers using a parallel scheme, whereas LoRA and DoRA are designed to tune the Q and V blocks, consistent with their original configurations. Furthermore, in Appendix B.2, we provide results demonstrating that the parallel control approach also outperforms a modified version of LoRA adapted to tune the MLP layers.

### 5.3. Scaling Up to Llama-2 and Llama-3

We extend our experiments to larger backbones, specifically LLaMA2-7B (Touvron et al., 2023) and LLaMA3-8B (Dubey et al., 2024), on the "Commonsense Benchmark". These experiments were first considered in DoRA (Liu et al., 2024), and we follow this pioneering work with an identical setup. Note that DoRA fine-tunes five components—Q, K, V, U, and D—for optimal performance, and we adopt this design. Specifically, we extend the control approach with two proposed solutions: the first combines control-based tuning for U and D with LoRA-based tuning for Q, K, and V, while the second adopts a unified double-control approach, tuning all five components using the control mechanism.

The results, as reported in Table 4, illustrate the performance of four fine-tuning methods on the LLaMA2-7B and LLaMA3-8B models. For the 7B model, the hybrid approach of Control(UD)+LoRA(QKV) outperforms DoRA in both accuracy and efficiency. With an average score of 80.0, it surpasses DoRA while significantly reducing 21G less memory and requiring almost half training time, demonstrating the advantages of strategically combining control mechanisms and LoRA for selective edge updates. On the

other hand, the double control approach achieves the lowest GPU memory usage and training time, consuming only 59% of the memory and 47% of the time required by DoRA, while delivering nearly identical performance (79.6 for double control versus 79.7 for DoRA). For the 8B model, the trends are consistent. The double control method achieves competitive performance (average score 85.3) while requiring only 67% of DoRA's memory and 51% of its training time.

It is important to highlight that the reduction in memory is not primarily due to the decrease in the number of parameters. For instance, halving the rank results in approximately a 50% reduction in parameters; however, the memory usage of DoRA only sees a marginal decrease of 0.256 GB. This indicates that the primary factor influencing memory consumption is not the parameter count but rather the intermediate states that need to be stored during the training process. By skipping these intermediate states, the control approaches maximally reduce the memory cost, while preserving or improving the overall performance.

### 5.4. Training 7B/8B Models on Nvidia-3090

Tuning 7B/8B models typically demands high-end GPUs, such as the Nvidia A100 with 80GB of memory. With the memory reduction provided by our approach, we investigate training these models on consumer-level hardware, specifically the Nvidia 3090 with 24GB memory. In contrast to the prior method (Zhao et al., 2024) on the 7B model, we refrain from using quantization in our approach: the pre-trained 7B/8B models are loaded in FP16, while the control weights remain in FP32.

By reducing the batch size to 4 and applying gradient accumulation with a single step, the proposed control-based approach exhibits only a minor performance degradation on the 7B model. For example, the double-control variant experiences a drop of just 0.7 points in accuracy. Despite this slight decrease, the model remains trainable on a single NVIDIA 3090 GPU, with a memory of only 20.656 GB. Notably, the performance degradation is even smaller on the 8B model, highlighting the scalability and practicality of the method on resource-constrained hardware.

Table 4: Comparison of algorithm performance on the Commonsense benchmark. Results show that the Control+LoRA and DoubleControl approaches obtain similar performance to DoRA with significant less GPU memories and computation time. Rank is set to 32 for all algorithms. † indicates numbers published in the original DoRA work.

| Model | Method | # of Params | GPU Memory | Training Time | BoolQ | PIQA | SIQA | HellaSwag | WinoGrande | ARC-e | ARC-c | OBQA | Avg |
|---|---|---|---|---|---|---|---|---|---|---|---|---|---|
| ChatGPT † | - | - | - | - | 73.1 | 85.4 | 68.5 | 78.5 | 66.1 | 89.8 | 79.9 | 74.8 | 77.0 |
| LLaMA2-7B | LoRA (QKVUD) † | 56.10 M | 44.204 GB | 8h37m | 69.8 | 79.9 | 79.5 | 83.6 | 82.6 | 79.8 | 64.7 | 81.0 | 77.6 |
| | DoRA (QKVUD) † | 56.98 M | 59.568 GB | 14h50m | 71.8 | **83.7** | 76.0 | 89.1 | 82.6 | **83.7** | 68.2 | **82.4** | 79.7 |
| | Control(UD)+LoRA(QKV) | 41.94 M | 38.556 GB | 7h36m | **73.0** | 83.5 | **79.5** | **89.7** | 82.6 | 82.9 | **68.6** | 80.4 | **80.0** |
| | DoubleControl (QKVUD) | 33.55 M | 35.214 GB | 6h58m | 72.3 | 82.5 | 79.2 | 89.1 | **83.1** | 83.0 | 68.5 | 79.0 | 79.6 |
| LLaMA3-8B | LoRA (QKVUD) † | 56.62 M | 55.040 GB | 9h33m | 70.8 | 85.2 | 79.9 | 91.7 | 84.3 | 84.2 | 71.2 | 79.0 | 80.8 |
| | DoRA (QKVUD) † | 57.41 M | 67.284 GB | 15h15m | 74.6 | **89.3** | 79.9 | 95.5 | 85.6 | 90.5 | **80.4** | 85.8 | 85.2 |
| | Control(UD)+LoRA(QKV) | 35.65 M | 48.550 GB | 8h11m | **75.7** | 87.9 | 80.4 | 95.5 | **86.3** | 90.6 | 79.8 | 86.2 | **85.3** |
| | DoubleControl (QKVUD) | 33.55 M | 45.316 GB | 7h44m | 74.1 | 87.8 | **80.7** | 95.5 | 86.0 | **90.8** | 80.0 | **87.8** | 85.3 |

Table 5: Training of 7B/8B models on Nvidia-3090. The batch size is set to 4 for the 7B model and 3 for the 8B model, with gradient accumulation set to 2. We omit comparison to LoRA and DoRA, as they are out-of-memory in this setting.

| Model | Method | # of Params | GPU Memory | Training Time | BoolQ | PIQA | SIQA | HellaSwag | WinoGrande | ARC-e | ARC-c | OBQA | Avg |
|---|---|---|---|---|---|---|---|---|---|---|---|---|---|
| LLaMA2-7B | Control(UD)+LoRA(QKV) | 41.94 M | 21.874 GB | 21h21m | 71.4 | 81.1 | 75.7 | 86.7 | 82.9 | 82.3 | 67.2 | 80.4 | 78.4 |
| | DoubleControl (QKVUD) | 33.55 M | 20.656 GB | 20h09m | 70.8 | 83.0 | 79.2 | 84.6 | 81.5 | 82.8 | 68.3 | 81.2 | 78.9 |
| LLaMA3-8B | Control(UD)+LoRA(QKV) | 35.65 M | 22.920 GB | 21h51m | 75.1 | 87.8 | 79.9 | 95.3 | 85.0 | 90.0 | 79.0 | 85.0 | 84.6 |
| | DoubleControl (QKVUD) | 33.55 M | 22.176 GB | 20h33m | 74.4 | 86.9 | 80.4 | 95.3 | 85.4 | 90.1 | 79.4 | 85.6 | 84.7 |

### 5.5. Ablation Studies on the Choice of Controlled Blocks

In this paper, we propose tuning an entire residual block by introducing an additional parallel edge in the computational graph. To motivate this design choice, we conduct an ablation study comparing different tuning granularities, including individual MLP layers, attention modules, and full residual blocks.

Table 6: Comparison of performance and GPU memory usage across different fine-tuning targets in ViT.

| | MLP | Attn | Full Block |
|---|---|---|---|
| **Performance** | $91.96_{\pm 0.05}$ | $91.93_{\pm 0.03}$ | $91.53_{\pm 0.11}$ |
| **GPU Memory** | 12.280 GB | 12.480 GB | 12.196 GB |

These results indicate that controlling either the MLP or attention layer individually leads to similar performance, with attention control slightly reducing accuracy and increasing GPU memory usage. In contrast, controlling the entire residual block results in a more notable 0.43% performance drop. This is likely because each ViT layer consists of two distinct residual blocks, and treating the entire block as a single unit reduces its effectiveness.

## 6. Conclusion and Future Works

In conclusion, this paper presents a shift from the traditional weight-centric fine-tuning approach to a state-based framework. Instead of focusing on weight adaptation, our method prioritizes adjusting forward states, with LoRA serving as a special case. By introducing parameterized perturbations to the computational graph, we can control entire residual blocks, significantly improving memory efficiency by reducing the need to store large intermediate states. Empirical results across multiple model architectures demonstrate that our approach further reduces memory usage and computation time, and also enhances performance.

While this paper focuses exclusively on edges associated with residual connections, this design choice represents just one possible solution of the broader state-based fine-tuning framework. Whether these edges are indeed the most effective or efficient targets for adaptation remains an open question. Future work may explore tuning edges beyond the original computational graph, including connections $(u, v) \notin E$, potentially unlocking new forms of interaction and control within the network. Additionally, we show that repositioning trainable matrices gives rise to a control-affine structure, though a comprehensive theoretical analysis of the resulting control system is left for future work.

## Acknowledgements

This research is supported by the National Research Foundation, Singapore, under the NRF fellowship (project No. NRF-NRFF13-2021-0005).

## Impact Statement

This paper presents work whose goal is to advance the field of machine learning, in particular the parameter-efficient fine-tuning. There are limited potential societal consequences of our work, none of which we feel must be specifically highlighted here.

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

# A. Proof for Theorems

## A.1. Proof of Theorem 4.1

*Proof.* $\bar{f} : x_0 \to x_T$ is given by

$$\bar{f}(x_0) = (W_T + R_T) \cdots (W_0 + R_0) x_t. \tag{13}$$

Notice that

$$\Delta W := (W_T + R_T) \cdots (W_0 + R_0) - W_T \cdots W_0 \tag{14}$$

$$= \sum_{t=1}^{T} \left( \prod_{i=t}^{T} (W_i + R_i) \prod_{i=1}^{t-1} W_i - \prod_{i=t+1}^{T} (W_i + R_i) \prod_{i=1}^{t} W_i \right) \tag{15}$$

$$= \sum_{t=1}^{T} \left[ \left( \prod_{i=t+1}^{T} (W_i + R_i) \right) R_t \left( \prod_{i=1}^{t-1} W_i \right) \right] \tag{16}$$

$$\tag{17}$$

Here, $\prod_{i=1}^{t} W_t := W_t W_{t-1} \cdots W_1$ denotes the left product of a sequence of matrices. Since

$$\text{rank} \left( \left( \prod_{i=t+1}^{T} (W_i + R_i) \right) R_t \left( \prod_{i=1}^{t-1} W_i \right) \right) \leqslant \text{rank}(R_t) \leqslant r_t, \tag{18}$$

we have $\text{rank}(\Delta W) \leqslant r_0 + \cdots + r_{T-1}$. Therefore, take $M = \Delta W$ gives the result. $\square$

## A.2. Proof of Theorem 4.2

*Proof.* Consider $\Delta W$ to be small, Taylor expansion of $F_{x_t}$ at zero gives:

$$F_{x_t}(\Delta W) = x_t + f_t((W_t + \Delta W_t) x_t) = x_t + f_t(W_t x_t) + \nabla f(x_t) \Delta W x_t + \mathcal{O}(\|\Delta W\|_2^2) \tag{19}$$

Therefore, $\nabla f(x_t) \Delta W x_t$ can only take values in the image space of $\nabla f(x_t)$, which is a proper subspace of $\mathbb{R}^d$ when $\nabla f(x_t)$ is singular.

Moreover, recall that

$$G_{x_t}(\Delta W_t) = x_t + f_t((W_t x_t) + \Delta W_t x_t, \tag{20}$$

it is obvious that $\Delta W_t x_t$ can take arbitrary value in $\mathbb{R}^d$ as long as $x_t$ is non-zero. $\square$

# B. Experiment Details

We present the details of our experimental setups in this section to provide the configurations and methodologies employed. This includes the specifications of the datasets, model architectures, and hyperparameter configurations, as well as the training strategies and evaluation protocols used in our experiments.

## B.1. Glue Benchmark with RoBERTa Models

The GLUE (General Language Understanding Evaluation) benchmark (Wang, 2018) is a widely used suite of natural language understanding tasks designed to evaluate the performance of machine learning models across a diverse range of linguistic challenges. It includes tasks such as sentence similarity (STS-B), natural language inference (MNLI, RTE, QNLI), sentiment analysis (SST-2), textual entailment (WNLI), question answering (QQP), and linguistic acceptability (CoLA). This dataset is widely utilized as a benchmark in prior studies on LoRA (Hu et al., 2021), serving as an evaluation framework for assessing the effectiveness of low-rank adaptation techniques.

For datasets with limited samples, such as MRPC, RTE, and STS-B, the initial LoRA paper employs a strategy of first pre-training models on the MNLI dataset, leveraging its larger size to improve performance, and subsequently selecting the best-performing checkpoint as the initialization for fine-tuning. While this approach can enhance downstream task performance, it introduces additional complexity to the experimental workflow and may pose challenges for reproducibility.

Following the simplified methodology proposed by (Wu et al., 2024), we omit this pre-training step, prioritizing a more straightforward and accessible experimental design.

We utilize the RoBERTa-base and RoBERTa-large models as the backbone architectures, incorporating LoRA, DoRA, or control modules, and directly fine-tuning these enhanced models on each dataset. This straightforward approach eliminates any need for additional pre-training steps. Generally, for datasets with a large number of samples, such as MNLI, the performance remains consistent (e.g., $87.5\%$ in the original paper, $87.57\%$ in Table 2). However, for smaller-scale datasets, such as MRPC or STS-B, we observe a slight decline in performance compared to previously reported results, likely due to the absence of specialized pre-training on MNLI, or the shrink for sequence length to 128 in our experiment. Despite this, the simplified workflow improves reproducibility while still maintaining similar performance across tasks.

We report the average performance over three random seeds (40, 41, and 42). For each run, the result is determined based on the best-performing epoch, selected according to the validation set performance. The rank of LoRA and DoRA is set as 8, and its alpha is set as 16. Since these methods need to tune both Q and V matrices, while Control only needs to tune one block, we set the rank of control to 16. This allows all algorithms to have similar parameters. Details of hyperparameter configurations are listed as follows.

| Model | Dataset | MNLI | SST-2 | MRPC | CoLA | QNLI | QQP | RTE | STS-B |
|---|---|---|---|---|---|---|---|---|---|
| | Optimizer | | | | AdamW | | | | |
| | WarmUp | | | | 0.06 | | | | |
| | Scheduler | | | | Linear | | | | |
| | Seq Length | | | | 128 | | | | |
| Rob$_{base}$ | Batch Size | 128 | 128 | 128 | 64 | 256 | 128 | 128 | 128 |
| | Epochs | 30 | 50 | 30 | 80 | 25 | 25 | 80 | 40 |
| | Learning Rate | 5e-4 | 5e-4 | 4e-4 | 3e-4 | 4e-4 | 5e-4 | 4e-4 | 4e-4 |
| Rob$_{large}$ | Batch Size | 32 | 64 | 32 | 32 | 32 | 32 | 64 | 32 |
| | Epochs | 10 | 20 | 30 | 20 | 10 | 20 | 20 | 10 |
| | Learning Rate | 3e-4 | 4e-4 | 3e-4 | 5e-4 | 3e-4 | 3e-4 | 4e-4 | 3e-4 |

Table 7: Hyperparameters on GLUE benchmark datasets.

## B.2. Additional Experiments on GLUE Benchmark

The control method primarily targets tuning the FFN block, whereas LoRA focuses on updating the Q and V matrices within the attention block. Our experiments reveal that the control method outperforms LoRA, particularly on datasets such as CoLA. To further investigate, we conduct an additional experiment where we apply LoRA to tune both layers of the FFN block.

| Method | Seed 40 | Seed 41 | Seed 42 | Average |
|---|---|---|---|---|
| LoRA | 61.92 | 63.93 | 60.59 | $62.15_{\pm 1.37}$ |
| Control | 65.36 | 64.84 | 65.77 | $65.32_{\pm 0.44}$ |

Table 8: Performance of LoRA and Control on tuning U and D matrices.

Specifically, we repeat this experiment on the CoLA dataset using random seeds ranging from 40 to 42 to ensure the consistency of the results.. The table 3 presents the results of two methods, LoRA and Control, evaluated across these three random seeds with their respective average performances. The results suggest that it is possible to control both MLP layers simultaneously, similar to the parallel control approach, rather than tuning each MLP layer individually.

## B.3. LLaMA-2 and LLaMA-3

The LLaMA2-7B (Touvron et al., 2023) and LLaMA3-8B (Dubey et al., 2024) models are part of the LLaMA (Large Language Model Meta AI) family, designed for efficient scaling in natural language understanding and generation tasks.

On the other hand, the commonsense reasoning tasks consist of 8 distinct sub-tasks, each with its own designated training and testing sets. In line with DoRA, we combine the training datasets from all 8 sub-tasks to form a unified training set. The pretrained models are first tuned on this combined training set, and then evaluations are carried out separately on the individual testing sets for each sub-task.

This experiment was first explored in DoRA (Liu et al., 2024), and we replicate the experimental setup as outlined in the original paper. In prior work with the RoBERTa model, tuning strategies were restricted to specific components: LoRA and DoRA primarily focus on adjusting the Q and V matrices, while the control mechanism targets the U and D matrices within the FFN block. However, for more complex architectures like LLaMA, it is generally advised (Liu et al., 2024) to tune all five components—Q, K, V, U, and D—for optimal model performance. To address this, we propose two extensions to the control approach: the first combines LoRA for tuning the Q, K, and V matrices with control-based tuning for the U and D matrices, while the second utilizes a unified double-control strategy, which tunes all five components through the control mechanism to enhance model flexibility and accuracy.

For the hyperparameter configuration, we use the AdamW optimizer, and the maximum sequence length is set to 256. Following DoRA, the training is conducted over a total of 3 epochs, with a batch size of 16 to balance computation and memory usage. On Nvidia-3090, the batch size shrinks to 4 for LLaMA2-7B and 3 for LLaMA3-8B model, with another round of gradient accumulation. For LoRA and DoRA, the rank is set to 32, and the alpha value is set to 64. In contrast, the rank for the Control approach is set to 64, since it tunes both U and D with one simple control unit.

### B.4. Sensitivity on the Ranks

We conduct a sensitivity analysis on the RTE dataset by varying the control rank r to assess its effect on GPU memory consumption, training time, and model accuracy.

Table 9: Effect of rank $r$ on GPU memory, training time, and accuracy.

| Rank (r) | GPU Memory | Training Time | Accuracy |
|---|---|---|---|
| 1 | 12.620 GB | 4m 58s | $73.05_{+0.34}$ |
| 4 | 12.622 GB | 4m 59s | $73.41_{+1.48}$ |
| 8 | 12.626 GB | 4m 59s | $74.37_{+0.59}$ |
| 16 | 12.634 GB | 4m 59s | $76.89_{+0.78}$ |
| 32 | 12.674 GB | 5m 00s | $77.17_{+0.34}$ |
| 64 | 12.724 GB | 5m 02s | $77.01_{+0.68}$ |

As the rank of the control parameters increases, GPU memory usage remains relatively stable, exhibiting only a slight increase. This stability stems from the use of low-rank matrices, which efficiently limit the memory overhead associated with storing backward gradients. In terms of accuracy, we observe consistent improvements up to a rank of 32, suggesting that increased rank enhances the expressiveness of the adaptation. However, at rank 64, the performance slightly declines compared to rank 32, indicating diminishing returns and potential overfitting beyond a certain threshold.

Next, we compare the performance of Control, LoRA, and DoRA across different ranks: At lower ranks (r=8), LoRA and

Table 10: Comparison of Control, LoRA, and DoRA across different ranks $r$.

| Method | Control | LoRA | DoRA |
|---|---|---|---|
| Accuracy ($r = 8$) | $74.37_{+0.59}$ | $74.84_{+1.48}$ | $74.97_{+1.45}$ |
| Accuracy ($r = 16$) | $76.89_{+0.78}$ | $75.79_{+1.57}$ | $76.05_{+1.67}$ |
| Accuracy ($r = 32$) | $77.17_{+0.34}$ | $75.97_{+2.07}$ | $76.65_{+1.19}$ |
| Accuracy ($r = 64$) | $77.01_{+0.68}$ | $75.93_{+1.78}$ | $76.78_{+2.01}$ |

DoRA achieve slightly higher accuracy than the control method, indicating that traditional weight-centric approaches may have an edge with limited parameter capacity. However, as the rank increases to 16 and beyond, the control method begins to outperform both LoRA and DoRA, demonstrating its ability to better leverage the increased expressiveness. Since raising

the rank to 16 or 32 results in only minimal increases in GPU memory usage and training time, we recommend adopting relatively higher ranks to maximize performance gains. Notably, this trend is consistent across other datasets such as CoLA and SST-2, highlighting the general effectiveness of the control-based fine-tuning approach.

