# OpenReview forum: "From Weight-Based to State-Based Fine-Tuning: Further Memory Reduction on LoRA with Parallel Control"
_ICML.cc/2025/Conference — ICML 2025 oral_

### Official Review · Reviewer_eRj7 · 2025-03-06

**Overall Recommendation:** 3

**Summary:**

This paper discussed the PEFT from a new view of control theory.  From control theory, a new State-Based Fine-Tuning (State-FT) is proposed, where the network is modeled as a graph with each edge representing weights and each node representing activations. Thus, any components such as MLP or a couple of layers can be viewed as a unified unit, whose output state (activations) can be modified by the trainable parameters/non-linear function. This method can reduce GPU memory since the middle activations that need to be saved for calculating gradient are reduced. Experiments prove this method's effectiveness.

## update after rebuttal

The authors have solved my concerns properly and I recommend accepting this paper.

**Claims And Evidence:**

Yes. There is clear and convincing evidence.

**Essential References Not Discussed:**

This paper has discussed related works.

**Experimental Designs Or Analyses:**

The experimental designs (RoBERTa, LLaMA2-7B, LLaMA2-8B, ) are fair.

**Methods And Evaluation Criteria:**

Yes.

**Other Comments Or Suggestions:**

No

**Other Strengths And Weaknesses:**

Strengths:

1. Clear writing and good motivation for this state-based FT method.
2. Good results.

Weaknesses:

1. Lack of clear discussion about the difference between the current work and the pioneering work (Zhang et al., 2024b).
2. Although the paper may seem complicated, the method is very simple. I think the previous part about control theory is more like an excessive theoretical embellishment.
3. To provide more insight, could you provide more experimental results if using other units? For example, using a total block (contain ATTN and FFN) as the unit.

**Questions For Authors:**

No

**Relation To Broader Scientific Literature:**

This paper mainly related to the PEFT area such as LORA and DoRA.

**Theoretical Claims:**

Yes. I have done my best to check the correctness of Theorem 4.1.

---

> ### Author Rebuttal · Authors · 2025-04-01
>
> Thank you for your thoughtful and positive comments. We would like to clarify on a few points to address the concerns raised.
>
> ### 1.Discussion about the difference between the current work and the pioneering work (Zhang et al., 2024b).
>
> We agree with the reviewer that it is important to clearly highlight the differences between our approach and previous work. To address this, we would like to include the following discussion in the revised version.
>
> In general, the differences between the current work and the pioneering work lie in two aspects.
>
> (1) **Different Aspects of Control Theory**. The previous work primarily focuses on the controllability aspect of control theory, aiming to understand how system parameters can be adjusted to influence system behavior. In contrast, our work introduces the concept of state-based tuning, drawing inspiration from feedback control systems. Rather than focusing on the controllability of the system, our method emphasizes how adjusting the system's state (as opposed to directly manipulating model weights) can lead to improved performance. This distinction highlights that these two papers target different areas of control theory.
>
>
> (2) **Memory and Computation Efficiency**. The most notable difference lies in the memory and computation efficiency. The prior approach enhances the LoRA method by introducing additional cross-patch attention mechanisms to modify the low-rank matrices. While this enhances the model's complexity, it also leads to increased memory consumption and training time due to the added operations and parameters. In contrast, our primary goal is to achieve memory efficiency by reducing the memory footprint, specifically by bypassing large intermediate states during model execution.
>
> ---
>
> ### 2. Control theory is like theoretical embellishment.
>
> We notice the reviewer’s concern regarding the role of control theory in our work and would like to clarify its necessity and impact.
>
> (1) **Bridging PEFT and State-Based Control**. While PEFT methods like LoRA primarily focus on tuning model weights, classical control theory is centered around adjusting system states, as seen in feedback control. These two perspectives may initially seem unrelated, but our key argument is that LoRA can be understood as a special case of state-based fine-tuning. This reframing is not just a theoretical exercise—it provides a new lens for understanding and designing PEFT algorithms. This allows our framework to establish a direct connection between PEFT and state-based control in deep NNs. This bridging alone is the key message that authors want to convey in the first part.
>
>
> (2) **Unlocking New Opportunities**. Control theory offers a wealth of results that could inspire and inform the study of PEFT algorithms. By framing LoRA within a control-theoretic perspective, we naturally arrive at several open research questions. For example, an essential question arises: Which states should we tune? This corresponds to selecting the appropriate edges on the computational graph, analogous to choosing control variables in feedback systems.  Many well-established results from control theory could potentially be applied here, but they have not been fully explored in the PEFT community.
>
> Thus, the integration of control theory is not mere embellishment; it serves as a foundation that opens new avenues for both theoretical and practical advancements. We will ensure that this motivation is articulated more clearly in the revised version.
>
> ---
> ### Distinguishing the Proposed Method from LoRA
>
> The second part of our paper provides a shift where we reposition the low-rank matrices. We acknowledge that our approach is simple and appears structurally similar to LoRA, yet the underlying formulation corresponds to distinct control frameworks.
>
> (1) **Different Forms of Control Systems**. From a control-theoretic standpoint, LoRA and DoRA can be viewed as *non-affine* control mechanisms:
>
> $\dot{x}(t) = f(x(t), u(t))$,
>
> where the control is deeply entangled with the model’s attention mechanisms. In contrast, our proposed parallel control formulation aligns more closely with *affine control*:
>
> $\dot{x}(t) = f(x(t)) + x(t) u(t)$,
>
> where the control component is decoupled from the function f. This distinction is crucial—affine and non-affine controls represent fundamentally different classes of control design, each with unique stability and efficiency properties. Our work leverages this difference to optimize memory consumption.
>
>
> (2) **Practicality of the Proposed Algorithm**.  Beyond above considerations, a key advantage of our method is its simplicity and ease of integration. By making minimal modifications to existing architectures, our approach remains practical and can be readily applied to various model designs.
>
> ---
>
> ### 3. More experimental results if using other units like a total block.
>
> Please kindly refer to our response to Reviewer 1 (hdej).

---

> > ### Comment · Reviewer_eRj7 · 2025-04-02
> >
> > Thank you for your rebuttal
> >
> > I still have some further questions after reading your rebuttal.
> >
> > 1. Which states should we tune? Could you show some examples by borrowing ideas from feedback systems? This would be interesting.
> >
> > 2. Affine and non-affine controls have different unique stability and efficiency properties. Could you explain these more clearly?

---

> > > ### Author Response · Authors · 2025-04-04
> > >
> > > These topic may slightly go beyond the scope of the current paper, but of scientific interest - we authors would be glad to engage in some discussions.
> > >
> > > ###  Examples by borrowing ideas from feedback systems
> > > This is indeed an interesting yet open question. Current matrix selection in PEFT largely guided by empirical studies. For instance, Section 7.1 of the LoRA paper empirically investigates which weight matrices should be adapted under a constrained parameter budget.
> > >
> > > Classical control theory may offer several principles that could provide insights into this problem:
> > >
> > > 1. **Controllability Analysis**: When tuning a subset of parameters, priority should be given to those with the greatest influence on the system’s ability to reach desired states. The *controllability Gramian* quantifies how effectively states can be controlled, potentially guiding the parameter selection for adaptation. Notably, the *network controllability* [4] focuses on identifying the minimum number of driver nodes, whose control is sufficient to fully control the system's dynamics. This should be relevant to efficient finetuning.
> > >
> > > 2. **Relative Gain Array (RGA)**: The RGA offers a normalized representation of the system’s gain matrix, highlighting the relative influence of each control variable on various outputs. Given linearization for finetuning model, this framework may help identify the most effective parameter subsets for model adaptation.
> > >
> > > ---
> > >
> > > ### Discussions on Affine and Non-Affine Control
> > > In classical control theory, researchers and engineers often **prefer control-affine systems** when possible because they're easier to analyze and implement.
> > >
> > > To see this, consider a simplified DC motor model:
> > >
> > > ẋ₁ = x₂
> > > ẋ₂ = -a·x₂ + b·u
> > >
> > > where x₁ = θ (position), x₂ = ω (angular velocity). This is a control-affine system. With state feedback control u = -k₁x₁ - k₂x₂, the closed-loop dynamics become:$
> > > A =
> > > \begin{bmatrix}
> > > 0 & 1; \\
> > > -b k_1 & -a - b k_2
> > > \end{bmatrix}
> > > $
> > >
> > > For stability, we may want poles at s = -p₁ and s = -p₂.
> > > This gives: k₁ = p₁p₂/b, and k₂ = (p₁+p₂-a)/b. This allows us to design control to reach the desired state in a **closed-form solution**.
> > >
> > > In contrast, a non-affine system such as ẋ₂ = -a·x₂ + b·tanh(u)
> > > requires *iterative* or *nonlinear* techniques, complicating stability analysis and tuning.
> > >
> > > #### the MPC Case
> > > Compared to non-affine control, affine control has more computational efficiency and analytical tractability. For instance, in model predictive control (MPC) for the dynamics
> > > $$
> > >     x_{n+1}=f(x_n)+g(x_n)u_n
> > > $$
> > > with the quadratic cost function:
> > > $$
> > > J=\sum_{k=0}^N \left[ (\bar x_{n+k}-x^{ref})^\top Q (\bar x_{n+k}-x^{ref})+u_{n+k}^\top R u_{n+k} \right].
> > > $$
> > > Here, $x^{ref}$ denotes the reference state. Since the system is affine in u, the cost function is **quadratic and convex** in $u$. In contrast, if the system is non-affine in $u$, the cost might be non-convex.
> > >
> > >
> > > #### Example from stability analysis
> > > Moreover, the stability analysis of control-affine systems is well-established in the literature[2], since the affine structure simplifies the construction of control Lyapunov and barrier functions. For example, the control Lyapunov function for a general control system $\dot x(t)=f(x(t), u(t))$ with control set $U$ is defined as a function $V>0$, such that for any $x$,
> > > $$
> > > \inf _{u\in U} \langle \nabla V(x), f(x,u) \rangle < 0.
> > > $$
> > > For general non-linear systems, such a condition is difficult to identify. However, for control-affine systems $\dot x(t)=f(x(t))+g(x(t))u(t)$, the condition becomes more concrete as
> > > $$
> > > \inf _{u\in U}\left[L_f V(x)+L_g V(x) u\right] < 0
> > > $$
> > > where the optimization problem is **linear in $u$**. Here $L_f V$ denotes the Lie derivatives of $V$ along $f$. Withproper linearization techniques, the Lyapunov function can be designed for many systems.
> > >
> > >
> > >
> > > #### When do we use non-affine control?
> > > Conversely, non-affine control is more difficult to analyze due to the non-linear dependence on control. However, they may provide more flexibility and potential benefits in complex systems.
> > >
> > > Non-affine control systems might be preferred when:
> > >
> > > * The physical system *inherently* has non-affine dynamics
> > > * More complex control behaviors are required or the application demands non-linear control responses.
> > >
> > > Finally, LoRA operates as a non-affine control mechanism, whereas our method follows an affine-control framework. This paper focuses on the memory reduction part, and the theoretical benefits need further study.
> > >
> > > ---
> > >
> > > [1] Model predictive control: theory, computation, and design (Vol. 2). Madison, WI: Nob Hill Publishing, 2017.
> > >
> > > [2] A survey on the control lyapunov function and control barrier function for nonlinear-affine control systems. IEEE/CAA Journal of Automatica Sinica, 2023.
> > >
> > > [3] Model predictive control for nonlinear affine systems based on the simplified dual neural network. In IEEE Control Applications,(CCA) & Intelligent Control, 2009.
> > >
> > > [4] Controllability of complex networks. Nature, 2011.

---

### Official Review · Reviewer_jhpo · 2025-03-11

**Overall Recommendation:** 3

**Summary:**

This paper presents a state-based fine-tuning framework, which can avoid storing large intermediate states during training. Empirical results show its effectiveness.

**Claims And Evidence:**

The claims made in the submission are supported by clear and convincing evidence.

**Essential References Not Discussed:**

n/a

**Experimental Designs Or Analyses:**

I have checked the soundness/validity of any experimental designs or analyses.

In Table 4, the comparison between ChatGPT and other baseline methods seems unfair, as there is no fine-tuning in ChatGPT.

**Methods And Evaluation Criteria:**

The proposed methods and/or evaluation criteria (e.g., benchmark datasets) make sense.

**Other Comments Or Suggestions:**

n/a

**Other Strengths And Weaknesses:**

Strengths:
1. This paper is well-written and easy to read. Figures 1, 2 clearly show the advantage of the proposed method.
2. The experiments of this paper validate the proposed method's efficient.

Weakness:
1. The contribution of this work is limited. The idea of state-based fine-tuning seems to apply the LoRA from the QKV matrix to the FFN/ATTN block, which appears to be an incremental extension of existing weight-based methods like LoRA. Moreover, the authors acknowledge that LoRA can be viewed as a special case of their framework (Section 3.3). This suggests that the proposed method is not fundamentally new but rather a generalization of existing techniques.
2. The empirical results, while showing some improvements in memory efficiency and training time, do not convincingly demonstrate the superiority of the proposed method. The performance gains are marginal (e.g., 0.12% improvement in accuracy on ViT, Table 1)
3. The paper lacks a strong theoretical foundation to justify the state-based tuning framework.  The connection between control theory and state-based tuning is superficial.

**Questions For Authors:**

n/a

**Relation To Broader Scientific Literature:**

The key contributions of the paper are related to the broader scientific literature in PEFT methods, such as LoRA and DoRA.

**Theoretical Claims:**

I have checked the correctness of any proofs for theoretical claims.

---

> ### Author Rebuttal · Authors · 2025-04-01
>
> Thank you for your valuable suggestion.
>
> ### 1. The idea of state-based fine-tuning seems to apply the LoRA from the QKV matrix to the FFN/ATTN block
>
> We understand the reviewer’s concern that our contribution may seem limited to proposing an algorithm, and we would like to clarify on this.
>
> First of all, our goal of proposing the State-FT method is to provide a **new viewpoint** to link the LoRA method with control approaches. While LoRA is typically framed as a low-rank fine-tuning method that modifies specific model weights, this contrasts sharply with classical control theory, which focuses on adjusting system states, as seen in feedback control. These two areas have largely developed independently. A central argument of our work is that LoRA can, in fact, be viewed as a special case of state-based fine-tuning. As noted by Reviewers hdej and uZCq, our framework aims to establish a direct connection between parameter-efficient fine-tuning and control formulations in deep neural networks. This is the key message we want to deliver when establishing the state-based FT.
>
> Such a viewpoint also paves the way for new algorithmic design. We propose to treat fine-tuning neural networks as modifications to its computational graph (DAGs), allowing us to rethink tuning as a structural modification of the network itself. This means we can introduce or remove specific edges within the computation graph based on task-specific requirements, rather than being limited to conventional weight adjustments.  For example, we can add edges that may not exist on the original network, to alter the computation flowmodifying the computation flow—something not captured by the original LoRA method. This viewpoint raises several **open questions** within the state-based tuning paradigm. For instance, *which states should be adjusted within the network, and how can we determine the optimal control strategy*? There are many well-established results from control theory, and we would expect them to be applied to finetuning research.
>
> Finally, we introduce an algorithm that tunes entire blocks as a unified unit. Our design philosophy is straightforward: **prioritize simplicity and effectiveness**.  By making the whole architecture simple and making minimal changes, we expect our approach remains broadly applicable across various network architectures.
> In the meanwhile, we would expect the algorithm to be effectively reduce the GPU memories, as demonstrated in the following experiments.
>
> In summary, we want to emphasize that our goals would extend beyond one specific algorithm—we aim to bridge weight-based fine-tuning with the well-established principles of state-based control theory. This is the primary contribution of this paper. This perspective not only offers a fresh way to interpret LoRA but also raises several open questions, paving the way for future exploration. We will make these points clearer in the updated version.
>
> ---
>
> ### 2. The empirical results show some improvements in memory efficiency and training time, but the performance gains are marginal as 0.12%.
>
> As our title suggests, the core objective of our method is to minimize memory consumption and computational overhead, making it an efficient solution. We demonstrate that our approach reduces memory usage from 18.010GB to 12.280GB on the ViT toy model, achieving a 31.8% reduction.
>
> The performance evaluations we include serve a specific purpose: to demonstrate that these efficiency gains do not come at the expense of model quality. We hope the reviewer can understand that the 0.12% difference is not intended to suggest a significant improvement over the baseline. Rather, its goal is to convey another message—that our method enhances efficiency without sacrificing effectiveness.
>
> ---
>
> ### 3. Strong theoretical foundation to justify the state-based tuning framework.
> Overall, we hope the reviewer can understand that this paper is not intended to be a theoretical paper. Instead, the analysis is mainly to support our claims. As alluded to earlier, the primary objective of our algorithm is to demonstrate that improvements in memory and computational efficiency do not come at the cost of model performance. Therefore, our analysis is centered on this key point.
>
> For linear cases, Theorem 4.1 establishes that the parallel control method retains the same expressive power as LoRA, provided both methods share the same total rank. This guarantees that adopting parallel control does not lead to a loss in expressiveness. For nonlinear cases, Theorem 4.2 further demonstrates that parallel control can offer greater adaptability, particularly in scenarios where the original model exhibits degeneracies. These results reinforce the robustness of our approach across different settings.
>
> If the reviewer has any additional concerns regarding necessary analyses, we would greatly appreciate the feedback and are willing to address them accordingly.

---

> > ### Comment · Reviewer_jhpo · 2025-04-02
> >
> > Thank you for your detailed explanations, which address most of my concerns. After reading the rebuttal, I acknowledge that using the control viewpoint offers new insights into LoRA. Accordingly, I will raise my score to 3.
> >
> > However, as reviewer eRj7 also noted, I still believe that further connecting control theory or methods to state-based tuning would strengthen the paper.

---

> > > ### Author Response · Authors · 2025-04-03
> > >
> > > Thank you, Reviewer jhpo! We will refine the control theory discussion for better clarity. Specifically, we will add a subsection to explicitly highlight the non-affine control property of LoRA, while emphasizing that our method follows an affine-control approach. We hope this distinction will more clearly differentiate our method from a control theory perspective.

---

### Official Review · Reviewer_uZCq · 2025-03-13

**Overall Recommendation:** 4

**Summary:**

This paper proposes a novel state-based fine-tuning framework named State-FT for parameter-efficient algorithms. The authors shift the focus from traditional weight-based adaptations (e.g., LoRA and its variants) to directly optimizing the model’s intermediate forward states. Based on the inspiration of the control theory, the proposed method referred to as state-based fine tuning (State-FT), directly modifies the intermediate states in the computation graph rather than adjusts the model’s weight matrices.
The authors demonstrate that State-FT significantly reduces GPU memory usage by compressing intermediate layers using ‘DoubleControl’ methods. Experiments show that this approach maintains or outperforms the GLUE benchmark and commonsense reasoning tasks. Furthermore, it scales to larger models such as LLaMA2-7B and LLaMA-8B, enabling fine-tuning on GPUs that usual consumers use (under 24GB) without any quantization methods.

**Claims And Evidence:**

The paper’s main claims are convincingly supported by comprehensive theoretical control theory bases and experimental results. The proposed state-based fine-tuning approach shows significant GPU memory reductions (approximately ~3GB on medium-sized models RoBERTa and up to 40$ reduction on large models such as LLaMA2-7B and LLaMA3-8B), as the authors claimed. Additionally, the experimental results consistently show competitive or higher accuracy performance compared to baselines such as LoRA and DoRA, supporting the claim of representational capacity.

However, one of the limitations is the lack of explicit comparisons to quantization-based methods (e.g., QLoRA), which also aim at memory reduction. This is one of the concerns that the proposed method is showing effective since it lacks the baselines. Furthermore, experimental validation for large-scale models focuses primarily on NLP and commonsense reasoning tasks, while only a limited CIFAR-100 evaluation is provided for vision tasks. This scope raises concerns about the method’s applicability across diverse domains.

**Essential References Not Discussed:**

AdaLoRA (Zhou et al., 2023), which is related to dynamic rank allocation, should be additionally referenced for the related works.

**Experimental Designs Or Analyses:**

For the experimental settings, the authors show toy examples (ViT) to large-size datasets (LLaMA2-7B, LLaMA3-8B), GLUE, and commonsense benchmarks. They analyzed a number of parameters, GPU memory, and training time for each experimental setting that could prove their method’s effectiveness.

**Methods And Evaluation Criteria:**

The method not only introduces a novel framework but also clearly justifies it, showing how the proposed method generalizes and explains the better representational capacity. All of the proposed frameworks (State-FT and double control) are well motivated and logically supported, making effective use of control theory concepts to reduce total GPU memory usage.

The evaluation involves standard benchmarks such as GLUE and commonsense reasoning datasets previously conducted in the DoRA. It provided GPU memory usage, training time, and parameter counts comparing their approach with the original LoRA and DoRA under the same settings.
However, for the vision domains, the evaluations were limited, showing only the toy example since the proposed method could not be found to be effective.

**Other Comments Or Suggestions:**

1.	The term ‘parallel control’ in Section 4.1 248 could confuse the readers that it refers to ‘parallel pipelining.’ Using the term ‘block’ in this case seems more appropriate, as the control unit is not working in parallel. In addition, the proposed method is mainly about ‘double control’, so the title seems to make sense ‘parallel control’ into ‘double control.’

2.	Adding the results (at least QLoRA) might strengthen this paper’s contribution. The quantization method was mainly raised to reduce memory efficient training and complement the performance degradation. If the performance degradation is severe, State-FT’s necessity might be significantly increased. If not, it might be better that State-FT could apply the QLoRA method independently, as DoRA already showed.

3.	As the author mentioned their work limitations, it might be comprehensive to provide how much slower they were in the inference phase, at least in the appendix.

**Other Strengths And Weaknesses:**

**Strengths**

1.	Provides novel aspects of the PEFT area that original works did not: As this paper mentions, the authors were inspired by the control theory, which supports new aspects and possibilities in the PEFT area. This novelty distinguishes it from previous PEFT methodologies and shows comparable results.

2.	Provides strong theoretical justifications: The authors successfully demonstrate how state-based perturbations generalize and enhance the representation capabilities of original LoRA methods. They support their methods both theoretically and experimentally.

3.	Clear justification and motiviation: The necessity of the method is well presented, particularly through detailed analyses breaking down GPU memory usage. The paper highlights critical memory bottlenecks in previous PEFT approaches, clearly motivating the need for their ‘double control’ method. In the end, they successfully show its practical effectiveness on consumer GPU hardware.

**Weaknesses**

1.	Lack of comparison with quantization-based methods: Although the paper mentions significantly reducing GPU memory usage, it omits direct comparisons with quantization-based fine-tuning techniques such as QLoRA or QALoRA. Since the paper focuses on memory efficiency, evaluating or discussing results against QLoRA seemed to be needed.

2.	The proposed method is not guaranteed to work in other domains, such as vision, since it only shows toy examples. Previous work (DoRA) conducted effectiveness with image-video-text understanding domains, so providing results from other domain tasks might increase the confidence of this paper.

3.	Limited analysis of parameter sensitivity. This paper lacks sensitivity studies regarding the impact of hyperparameter settings (e.g., rank). Including sensitivity experiments could strengthen the paper’s framework.

**Questions For Authors:**

1.	Could the authors show the results, including the QLoRA and QA-LoRA quantization methods? In addition, could State-FT use the QLoRA method independently?

2.	Could the authors provide a sensitivity study of changing rank settings?

3.	The proposed method seems to be more efficient for large models such as LLaMA-13B or above. Could the authors provide results for larger models that could explain this method’s scalability?

4.	The results were only shown for the vision tasks for a small dataset (CIFAR-100). Could the proposed method be adopted for the Image-Video-Text understanding domains that use model backbone VL-BART, such as VQA, GAT, NVLR, and COCO Caption tasks? In the commonsense reasoning tasks, the performance seemed to be competitive with the existing method, but the question remains whether State-FT could remain accurate on multi-modality fine-tuning tasks, as DoRA has already shown. It might strengthen the paper if the state-FT works on even the multi-modality domains.

**Relation To Broader Scientific Literature:**

This paper provides new aspects over what the original parameter-efficient training tries to approach (weight-matrices approaches). In addition, they reduce the memory issues without any quantization schemes that were typically conducted for memory reduction. State-FT contributes to PEFT (parameter-efficient training) and memory efficiency areas such as QLoRA or parameter-sharing, while the work more directly addresses activation memory.

**Theoretical Claims:**

Section 3.3 of State-FT provides that the LoRA can be considered a special case of the proposed framework. There seem to be no issues with the theoretical claims.

---

> ### Author Rebuttal · Authors · 2025-04-01
>
> We sincerely thank the reviewer for your positive and insightful feedback.
>
> ### 1. Quantization Method like QLoRA/QA-LoRA. Can State-FT use QLoRA method independently?
>
> Yes, the State-FT method can independently leverage the QLoRA/QA-LoRA approach. GPU memory usage mainly arises from three sources: model weights, forward states, and backward gradients. QLoRA and QA-LoRA reduce memory by quantizing *model weights* and using low-rank matrices to minimize *backward gradients*. In contrast, State-FT focuses on reducing *forward states* and *backward gradients*. These methods are complementary, addressing different sources of memory cost.
>
> State-FT can be combined with the QLoRA method. This allows control to be operated on a quantization model, hence reducing memory consumption across all three categories. To validate this, we evaluate on the commonsense dataset with 8-bit LLaMa3-8B model.
>
> ||GPU Memory|Training Time|Accuracy|
> |-|-|-|-|
> |Double-Control|22.176 GB|20h33mins|84.7|
> |Q-Double-Control|17.796 GB|39h20mins|84.3|
>
> As shown, the use of QLoRA further reduces the memory consumption of the double-control approach by an additional 4.38 GB. However, this reduction in memory comes with trade-offs: training time increases by 91.40% and accuracy drops by 0.4%.
>
> We further investigated the effects of quantization with 4-bit methods on the RTE dataset.
>
> ||GPU Memory|Training Time|Accuracy|
> |-|-|-|-|
> |Control|12.634 GB|4min59s|$76.89 \pm 0.78$|
> |Q-Control|4.548 GB|9min03s|$73.41 \pm 1.36$|
> |Q-LoRA|4.688 GB|9min36s|$71.84 \pm 2.23$|
> |Q-DoRA|4.742 GB|11min03s|$71.00 \pm 0.94$|
>
> (1) As shown above, 4-bit quantization techniques effectively reduce GPU memory consumption. (2) Like its full-precision counterpart, Q-Control offers slight gains in memory, training time, and accuracy by reducing forward state memory, though overall savings remain modest. (3) Similar to 8-bit, this reduction comes at a cost—Q-Control incurs a 3.48% accuracy drop and an 81.61% increase in training time on the RTE dataset.
>
> In summary, the proposed state-FT can also benefit from methods like QLoRA. Quantization can effectively further reduce memory consumption for control and other PEFT methods, but comes with trade-offs, such as increased training time and slight performance degradation.
>
> ---
>
> ### 2. Sensitivity Study of Rank Settings.
>
> We conduct a sensitivity analysis on the RTE dataset by varying the control rank to evaluate its impact on GPU memory usage, training time, and accuracy.
>
> ||GPU Memory|Training Time|Accuracy|
> |-|-|-|-|
> |r=1|12.620 GB|4min58s|$73.05 \pm 0.34$|
> |r=4|12.622 GB|4min59s|$73.41 \pm 1.48$|
> |r=8|12.626 GB|4min59s|$74.37 \pm 0.59$|
> |r=16|12.634 GB|4min59s|$76.89 \pm 0.78$|
> |r=32|12.674 GB|5min00s|$77.17 \pm 0.34$|
> |r=64|12.724 GB|5min02s|$77.01 \pm 0.68$|
>
> As the rank of the control parameters increases, GPU memory usage remains relatively stable, with only a slight rise. This is due to the use of low-rank matrices, which help reduce the memory consumption of backward gradients. For accuracy, we observe a steady improvement up to a rank of 32. However, at a rank of 64, performance slightly declines compared to rank 32, indicating diminishing gains beyond a certain threshold.
>
> Next, we compare the performance of Control, LoRA, and DoRA across different ranks:
>
> ||Control|LoRA|DoRA|
> |-|-|-|-|
> |Accuracy(r=8)|$74.37 \pm 0.59$|$74.84 \pm 1.48$|$74.97 \pm 1.45$|
> |Accuracy(r=16)|$76.89 \pm 0.78$|$75.79 \pm 1.57$|$76.05 \pm 1.67$|
> |Accuracy(r=32)|$77.17 \pm 0.34$|$75.97 \pm 2.07$|$76.65 \pm 1.19$|
> |Accuracy(r=64)|$77.01 \pm 0.68$|$75.93 \pm 1.78$|$76.78 \pm 2.01$|
>
> (1) At lower ranks (r=8), LoRA/DoRA achieve higher accuracy than the control method. (2) As the rank increases (r=16 and above), the control method begins to outperform both LoRA and DoRA. (3)  Given that increasing rank to 16 or 32 has a minimal impact on training time and memory usage, we recommend using a relatively high rank to maximize performance. (4) This trend is also observed on CoLA and SST-2 datasets.
>
> ---
>
> ### 3. Large Models like 13B, and Different Domains
>
> We expect State-FT to achieve similar memory reductions on larger models like 13B. Training these models may take several days or even weeks, exceeding the rebuttal period. We are actively working on these experiments.
>
> Below, we present preliminary results from a multi-task evaluation with VL-BART:
>
> |Method|VQA|GQA|NVLR|COCO|Avg|
> |-|-|-|-|-|-|
> |FT|66.9|56.7|73.7|112.0|77.3|
> |LoRA|65.2|53.6|71.9|115.3|76.5|
> |DoRA|65.8|54.7|73.1|115.9|77.4|
> |Control|65.9|55.1|72.7|115.9|77.4|
>
> In terms of efficiency, the control method achieves a 31.6% reduction in memory usage and a 65.3% reduction in computation time compared to DoRA.
>
> ---
>
> ### 4. Essential References Not Discussed
> We notice [1] is a relevant paper and will include it in our reference list.
>
> ---
> [1]AdaLoRA: Adaptive Budget Allocation for Parameter-Efficient Fine-Tuning. ICLR 2023.

---

> > ### Comment · Reviewer_uZCq · 2025-04-07
> >
> > The authors have provided adequate responses to my concerns.
> > The results now seem to be more solid.
> >
> > I spent considerable time on whether to raise my score to a 4.
> > My concern comes from the fact that the proposed method essentially adds another path to the model, even though it is supported by a solid derivation.
> > However, since simple solutions that perform as well as more complex ones deserve recognition, I have decided to adjust my score to a 4.

---

> > > ### Author Response · Authors · 2025-04-07
> > >
> > > Thank you sincerely for your thoughtful consideration and for acknowledging our responses and the improvements made to the paper. Your feedback has been both encouraging and instrumental in helping us strengthen our work. We’re glad that the simplicity and effectiveness of our approach came through, and we truly appreciate the time and care you invested in the evaluation. Thank you once again for your valuable input!

---

### Official Review · Reviewer_hdej · 2025-03-14

**Overall Recommendation:** 4

**Summary:**

This paper proposes a new state-based fine-tuning framework that allows tuning entire residual blocks or multiple sequential sub-layers instead of adding adapters to each layer. The method significantly reduces the memory footprint by avoiding the storage of large intermediate activations while maintaining fine-tuning performance.

**Claims And Evidence:**

Overall, authors' claims are backed up by the provided experiments, which appear consistent with their theoretical motivation. More precisely, the experimental results support the paper’s claim that state-centric fine-tuning is more efficient and has same performance as weight-centric fine-tuning.

**Essential References Not Discussed:**

The paper covers the main PEFT references (LoRA, DoRA, and adapter/prompt-tuning families) and also cites relevant control-theoretic works. If anything, it might be beneficial to discuss more systematically how these “state-based” methods compare to existing large-activation checkpointing or gradient checkpointing strategies, which also address memory usage. Such a comparison would clarify the practical trade-offs.

**Ethical Review Concerns:**

No ethics concerns.

**Experimental Designs Or Analyses:**

Yes. No major issues. The experiments cover both small-scale (CIFAR-100 on ViT) and medium-scale (GLUE tasks on RoBERTa) fine-tuning, as well as large-scale LLaMA2-7B/ LLaMA3-8B models for common-sense QA tasks. The study ablates memory usage, training speed, and final performance scores. These are well-chosen metrics for a new PEFT algorithm.
One potential limitation is that each dataset is evaluated with a specific set of hyperparameters. It would be helpful to see more ablation on rank choices or to confirm that the approach remains stable with smaller ranks on more difficult tasks.

**Methods And Evaluation Criteria:**

Yes, they do.
In terms of methods, the paper’s core contribution is to replace weight-based LoRA with a “control-based” injection of low-rank updates in the forward pass. Instead of decomposing weight updates in Q/K/V or feed-forward layers, the method combines entire residual (or multi-layer) blocks into one function and introduces a separate control path that is likewise parameterized by low-rank matrices.
In terms of evaluation criteria, the authors rely on standard accuracy metrics for classification and QA tasks (e.g., MNLI accuracy, STS-B Pearson correlation, CoLA Matthew’s correlation, etc.), as well as training resource metrics such as GPU memory consumption and wall-clock training time. These metrics are appropriate for testing whether a method is truly more parameter- or memory-efficient.

**Other Comments Or Suggestions:**

Including a clear, step-by-step pseudocode for the “DoubleControl” approach in the Supplementary can help readers adopt the technique.

**Other Strengths And Weaknesses:**

Strengths:
- Comprehensive and strong experimental results.
- Significant memory savings allow training 7B/8B models on a single 24GB GPU (Nvidia 3090).
- The conceptual link between weight updates and states/controls may inspire further research.

Weaknesses:
- Merging weights or adapters from multiple “control edges” can be more complicated in practice than with purely weight-centric methods. The authors do note an increase in inference time, though they characterize it as modest.
- More ablation and clarity on how exactly “skipping” big intermediate states is implemented (particularly for widely used frameworks like PyTorch) would strengthen the practical dimension of the paper.

**Questions For Authors:**

1. How sensitive is the method to which blocks get controlled? Have the authors tried controlling only a portion of the MLP layers vs. controlling entire multi-head attention and feed-forward blocks together?
2. Can the authors elaborate on any overhead or complexity added during inference when “control edges” are introduced? Do you anticipate issues in deployment on standard inference platforms?
3. Is it feasible to combine two sets of “control edges” learned on different tasks? If so, how would that composition look?
4. Could gradient checkpointing or other memory-reduction approaches be integrated into this method to push memory usage down even further?

**Relation To Broader Scientific Literature:**

The paper is a novel connection between parameter-efficient fine-tuning and optimal control formulation for deep neural networks.
The paper positions itself as a generalization of LoRA in the context of control theory, a line of reasoning that has emerged in prior work linking neural networks to ODEs or closed-loop control systems. This is a valuable perspective The discussion of other memory-efficient techniques (like QLoRA or other quantization-based PEFT methods) is comparatively brief, but the authors’ emphasis is on building upon LoRA-style low-rank methods, so this is understandable.

**Theoretical Claims:**

Yes. I checked proofs of theorems 4.1 and 4.2 (guarantee of parallel control method performance). No major issues.

---

> ### Author Rebuttal · Authors · 2025-04-01
>
> We sincerely thank reviewer for the insightful and constructive feedback.
>
> ### 1. Sensitivity to the Choice of Controlled Blocks
> Controlling either the MLP or attention layer yields comparable performance, as demonstrated by the ViT model results:
>
> ||MLP|Attn|Full Block|
> |-|-|-|-|
> |Performance|$91.96\pm0.05$|$91.93\pm0.03$|$91.53\pm0.11$|
> |GPU Memory|12.280G|12.480G|12.196G|
>
> These results indicate that controlling either the MLP or attention layer individually leads to similar performance, with attention control slightly reducing accuracy and increasing GPU memory usage. In contrast, controlling the entire residual block results in a more notable 0.43% performance drop. This is likely because each ViT layer consists of two distinct residual blocks, and treating the entire block as a single unit reduces its effectiveness.
>
> To further examine the effects of tuning different portions of the MLP, we selectively tune either the first six or the last six layers of the ViT model while doubling the rank to maintain the same parameter number.
>
> ||First 6 Layers|Last 6 Layers|All 12 Layers|
> |-|-|-|-|
> |Performance|$91.14\pm0.03$|$87.23\pm0.11$|$91.96\pm0.05$|
> |GPU Memory|11.876G|7.180G|12.280G|
>
> These results highlight that selectively tuning a subset of the MLP layers leads to a more pronounced drop in accuracy. Specifically, tuning only the first six layers results in a 0.82% accuracy reduction. On the other hand, tuning only the last six layers leads to a more substantial memory reduction, as forward states for the first six layers no longer need to be stored. However, this comes at the cost of a significant 4.73% accuracy drop.
>
> While these experiments are conducted on the ViT model, we observe a similar trend across other architectures. These ablation studies will be added to the appendix.
>
> ---
>
> ### 2. Combining two sets of control edges.
> If two sets of edges share the same starting and ending nodes on a pretrained model, their combination can be conducted as existing LoRA merging techniques, such as ZipLoRA [1], K-LoRA [2] and SVD methods [3-4]. For example, given two sets of additional edges with weights $\Delta  W_1$ and $\Delta  W_2$, we can apply a similar approach as [1] and optimize $\ell_{\text{merge}}$ with
> $$\Delta W = v_1 \circ \Delta W_1 + v_2 \circ \Delta W_2,$$
> where $v_1$ and $v_2$ are vectors, and $\circ$ denotes element-wise multiplication, such that the j-th column of $\Delta W$ is scaled by the j-th element of v. The purpose of this lightweight tuning is to mitigate potential conflicts between different sets of low-rank matrices.
>
> If the two edges have different starting and ending nodes, they can be incorporated into the computation graph independently. However, two key considerations must be addressed. (1) Preserving the Directed Acyclic Graph (DAG): It is essential to ensure that the newly introduced edges do not violate the DAG structure. For example, if the edges include both $A \to B$ and $B \to A$, a cycle would be introduced, requiring structural trimming. (2) Resolving Conflicts: Different sets of edges may result in conflicting weight updates. To optimize performance, lightweight tuning remains a recommended step.
>
> ---
>
> ### 3. Compatibility with Gradient Checkpointing and Memory-Reduction Methods.
> Yes, we use gradient checkpointing when training 7B/8B models to reduce memory. For other methods like quantization, please kindly refer to our response to Reviewer 2 (uZCq).
>
> ---
>
> ### 4. Overhead During Inference and Deployment on Standard Inference Platforms.
>
> Extra control edges may lead to a slight increase in inference time. In the worst-case scenario, such as the double-control approach with two extra edges, our experiments show that the extra inference time is around $2.1\% \pm 0.3\%$ extra inference time for RoBERTa-base, and $4.6\% \pm 0.5\%$ for the LLaMA2-7B model.
>
> The introduction of control edges primarily impacts the forward pass by modifying the computation graph. Despite this, our method remains compatible with standard inference platforms like HuggingFace Transformers and Accelerators. While certain platform-specific optimizations, such as quantization, may require some adaptation, we do not anticipate significant changes.
>
> ---
>
> ### 5. Pseudocode and How to Implement Skipping Intermediate State.
> In general, steps to skip big intermediate states include:(1) Inherit the original layer and insert low-rank matrices; (2) Define a control function to compute the controlled state $x_c$; (3) Add $x_c$ to the original output.
>
> We will include the pseudocode in the appendix to provide further clarity and would like to thank you once again for this valuable suggestion.
>
> ---
> [1] ZipLoRA: Any Subject in Any Style by Effectively Merging LoRAs. ECCV 2024.
>
> [2] K-LoRA: Unlocking Training-Free Fusion of Any Subject and Style LoRAs. ArXiv 2025.
>
> [3] Task Singular Vectors: Reducing Task Interference in Model Merging. Arxiv 2024.
>
> [4] Model merging with SVD to tie the Knots. ICLR 2025.

---

### Decision · Program_Chairs · 2025-05-01

**Decision:**

Accept (oral)

**Comment:**

The paper proposes State-FT, a novel state-based parameter-efficient fine-tuning (PEFT) framework based on control-theory.  In contrast to weight-centric PEFT methods, i.e., LoRA and its variants, State-FT directly tunes the intermediate states of the computation graph rather than learning weight adapters.  Importantly, State-FT (which is also shown to generalize LoRA) avoids storing large intermediate activations while maintaining PEFT performance.  For attention-based LLMs, the authors demonstrate the efficacy of State-FT by introducing *parallel control*--wherein feed-forward network (FFN) blocks are fine-tuned--and *double control*--wherein both FFN and attention blocks are fine-tuned.  The authors show the efficacy of their framework across various models/sizes (ViT, RoBERTa, Llama-2-7B, Llama-3-8B), standard NLP tasks (e.g., GLUE and common-sense reasoning), and a toy vision example based on CIFAR-100.  Compared to LoRA/DoRA across the presented evaluations, State-FT methods are shown to significantly reduce GPU memory consumption and wall-clock time without degrading FT performance.

Reviewers agree that this approach is an interesting/novel addition to the long list of existing PEFT methods (while also noting the limitations of the vision results, which could be addressed in follow up works).  Reviewer uZCq noted the lack of comparison to QLoRA as a major concern.  However, the authors did well to show that State-FT and QLoRA are complimentary, effectively addressing these concerns during the rebuttal.  Further discussion included clarifications on the framing of existing PEFT methods in the context of control theory, which the authors showed offers a fresh take on PEFT and generalizes existing approaches.